# Leveraging LLM-based Multi-Agent Collaboration to Enhance Embodied Agents' Reasoning Capabilities for Solving Text-based Tasks in Human-populated Environments

**Nan Sun**
Department of Computer Science and Technology
Tsinghua University
sunn24@mails.tsinghua.edu.cn

**Chengming Shi**
Institute for Interdisciplinary Information Sciences
Tsinghua University
scm23@mails.tsinghua.edu.cn

**Yuwen Dong**
Department of Automation
Tsinghua University
dyw23@mails.tsinghua.edu.cn

## 1 Introduction

With the rapid advancement of artificial intelligence, the demand for intelligent robotic systems capable of interacting and assisting humans in real-world environments has surged. These systems are expected to interpret human instructions and perform tasks autonomously. However, existing service robots and virtual assistants still face significant challenges in dynamic reasoning, flexible task execution, and human interaction. They struggle to execute tasks as fluently as a human assistant, especially when text-based instructions are involved.

Recent developments in Large Language Models have shown promising results in overcoming some of these limitations by improving reasoning capabilities and enabling more natural language interactions. LLM-based agents have demonstrated significant performance in various tasks, including GUI-based operations, question answering, image description, and even complex embodied tasks such as robot navigation and mechanical arm manipulation. However, robots remain limited in their physical capabilities when it comes to handling complex embodied tasks, and agents still struggle to interact flexibly with humans for supplementary information or additional assistance.

This proposal explores the use of an LLM-based Multi-Agent Reasoning Framework designed to enhance embodied agents' ability to execute text-based tasks in human-populated environments. Our aim is to develop a collaborative system where LLM-driven agents assist robots in processing human instructions more effectively, while also seeking human assistance when necessary, ensuring task completion with minimal human intervention in domestic environments such as offices.

## 2 Problem Statement

Even though LLM-based agents have shown promising results in handling various tasks, robots still lack the flexibility and reasoning needed for efficient task execution. The problem becomes particularly evident in environments such as offices, where robots must perform a variety of tasks based on human instructions and collaborate with humans. Previous embodied agents face a number of constraints when executing tasks in real-world environments, particularly when these tasks are involved with other humans. Robots often struggle with:

Preprint. Under review.

- **Dynamic reasoning**: Inability to adjust to new or unexpected variables in real-time, e.g., when a robot goes to find a person for a signature but the person is not present or he does not have a pen.
- **Limited physical abilities**: Failure to complete complex instructions that involve delicate environmental interaction, e.g., fetching a cup of hot water from a water dispenser or purchasing a drink from a vending machine.
- **Human interaction**: Difficulty in seeking assistance when task completion is challenging due to physical or operational constraints.

The challenges are difficult to effectively solve under the single-agent architecture of existing work. This is due to the overly long sequential progress of solving a real-world task which limits performance. Therefore, It is essential to design a reasoning framework that leverages LLM-based multi-agent collaboration to overcome these limitations. Such a system would help robots better understand and execute text-based instructions, while also intelligently seeking human assistance when necessary.

## 3   Objectives

By integrating LLM-based multi-agent reasoning capabilities, robots are expected to gain improved flexibility and adaptability in real-world scenarios. Such a framework will enable robots to autonomously handle a wider range of tasks while minimizing the need for constant human supervision and intervention.

The objectives of this research are as follows:

1. **To develop an LLM-based multi-agent reasoning framework** capable of enhancing robots' ability to understand and execute text-based instructions in complex, human-populated environments. This will involve integrating multi-agent collaboration to enable robots to process dynamic tasks and adapt to real-time changes in their operational environment.
2. **To ensure that robots can efficiently seek human assistance** when necessary, while autonomously handling as many tasks as possible to minimize human intervention.
3. **To validate the proposed framework** in office or indoor scenarios, demonstrating its effectiveness in addressing day-to-day tasks and its ability to interact with humans for assistance when needed.

## 4   Preliminary Literature Review

To study scenarios involving multi-person interaction, we primarily chose to use mobile robots as the embodiment for our research on autonomous embodied agents.

### 4.1   Mobile Robots in Human-Populated Environments

The use of mobile robots in environments populated by humans has emerged as a key area of research within robotics and embodied AI. Initially, studies concentrated on robots operating in structured environments with limited human interaction. However, as demand for robots in more dynamic and unpredictable settings has grown, research has increasingly focused on improving adaptability and enhancing human-robot collaboration. Chung et al. [1] explored how mobile robots can autonomously collect and transmit environmental data to support human activities. Various researchers, such as Zhang et al. [2], Trautman and Krause [3], Truong and Ngo [4], Trautman et al. [5], examined robust navigation strategies for mobile robots functioning in complex, human-centered environments. In addition, Liang et al. [6] introduced a method enabling service robots to determine humans' dynamic locations through dialogue processing. Triebel et al. [7] developed a system for robots to sense, learn, and model human social behaviors, allowing them to make appropriate real-time decisions in their interactions.

### 4.2   LLM-Based Multi-Agent Systems

The field of multi-agent systems has seen significant advancements in recent years, particularly with the rise of large language models (LLMs) [8]. A multi-agent framework was employed by Wang et al.

[9, 10], Ma et al. [11], Cheng et al. [12], Tan et al. [13], Zhang et al. [14] to manage tasks related to GUI operations on smart devices. Similarly, Chan et al. [15] utilized a multi-agent approach to autonomously assess and discuss the quality of generated responses. Moreover, the evaluation of LLMs within multi-agent systems has been a focus for several researchers, including Abdelnabi et al. [16, 17]. These systems have also been applied extensively in communication scenarios between agents and humans, aimed at gathering detailed information [18, 19, 20]. Chen et al. [21] investigated how cyber agents from different networks could collaborate and share intelligence to enhance performance in diverse systems.

## 5   Methodology

This study will employ the following research methods:

1. **Multi-Agent Collaboration Framework Design:**
   - We will design a multi-agent reasoning framework in which LLM-based agents collaborate with each other. The LLM agents will handle complex text interpretation and dynamic reasoning to guide the robot in task execution and environmental interaction. When the robot encounters limitations in its physical capabilities, it will proactively seek assistance from human collaborators.

2. **Simulated and Real-world Testing:**
   - The framework will be validated in both simulated and real-world office environments. Robots will be tasked with executing various text-based instructions, such as object retrieval, navigation, and simple mechanical tasks. The system's performance will be evaluated mainly based on task completion rate and the extent of human intervention required.

3. **Evaluation Metrics:**
   - The system will be evaluated on its task success rate, collaborative efficiency with humans, and the reduction in human assistance required for task completion.

4. **Iterative Improvement:**
   - Feedback from humans and real-world testing will be used to refine the reasoning framework, improving its ability to handle increasingly complex tasks with minimal human intervention.

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
