# OpenReview forum: "【Proposal】Leveraging LLM-based Multi-Agent Collaboration to Enhance Embodied Agents’ Reasoning Capabilities for Solving Text-based Tasks in Human-populated Environments"
_tsinghua.edu.cn/THU/2024/Fall/AML — THU 2024 Fall AML Submission_

### Official Review · ~Liutao7 · 2024-11-06
**An Innovative and Promising Multi-Agent Collaboration Framework**

**Rating:** 9
**Confidence:** 4

**Review:**

This proposal presents a multi-agent collaboration framework that utilizes  LLMs  to enhance the reasoning capabilities of embodied agents for executing text-based tasks in human environments. I think: 1. The proposal has good integrity; 2. The proposal is highly creative, providing a solution to address the limitations of embodied agents in dynamic reasoning, physical capabilities, and human-robot interaction; 3. The workload is reasonable.
Areas for improvement: The implementation plan and evaluation criteria need further refinement.

---

### Official Review · ~Daniel_Wang4 · 2024-11-06
**Promising Framework for Enhancing Embodied Agents**

**Rating:** 9
**Confidence:** 3

**Review:**

This proposal presents a creative and well-rounded approach, using LLM-powered multi-agent collaboration to improve how embodied agents handle text-based tasks in human environments. It’s innovative in tackling the real challenges these agents face, like adapting to unexpected changes, handling physical tasks, and interacting effectively with people.

However, there are areas that could use a bit more clarity. The methodology feels a bit vague in parts, leaving questions about how exactly the multi-agent system will work in complex, real-world situations. Additionally, the evaluation criteria could be more detailed to make it easier to see how success will be measured across various tasks and environments.

Overall, the proposal is thoughtfully put together and has strong potential to advance the effectiveness of these agents in practical settings.

---

### Official Review · ~Un_Lok_Chen1 · 2024-11-08
**An LLM-based Multi-Agent Framework for Complex Robotic Tasks**

**Rating:** 8
**Confidence:** 4

**Review:**

Summary:

This project proposal aims to explore an innovative multi-agent framework powered by LLMs to enable robots to handle dynamic tasks that require more reasoning abilities and interact with humans and external environment. It also outlines a preliminary plan to iteratively implement and validate the framework in simulated and real-world scenarios such as an office.

Pros:

1. The proposal has well presented the problem to be solved and the motivation behind their objectives. The research problem is considered critical and concrete.

2. The language use is professional and clear.

3. The structure of the proposal is reasonable enough to guide the readers from problem introduction to related works and at last the methodology. Reading experience is smooth.

Cons:

A. Major issues:

1) There are at least 3 goals enlisted in the proposal: 1) to incorporate LLM to allow for human interaction with text-based instruction; 2) to develop reasoning abilities of the robot for dynamic tasks with LLM-based agents; 3)  to implement mechanisms for the robot to seek for human assistance actively. Additionally, these goals are supposed to be tested on robots in real complex scenes. The objective of the work may be too ambitious and the workload may be too high for a course project.

B. Minor issues:

1) If possible, consider adding more references to previous works to support the claims presented in the Introduction and Problem Statement sections.

2) Please comply with the maximum length requirement in the proposal guideline (2 pages).

3) Consider elaborate more on what specific techniques may be used in the Multi-Agent Collaboration Framework.

4) Are there any specific baseline methods on this task to compare with?

---

### Official Review · ~Ruitao_Jing1 · 2024-11-08
**A Novel LLM-Driven Multi-Agent Robotics Frameworks in Human-Centric Environments**

**Rating:** 9
**Confidence:** 3

**Review:**

This paper presents a novel framework leveraging Large Language Models (LLMs) to enhance the capabilities of multi-agent robotic systems in executing complex tasks within human-centric environments. The framework aims to address the current deficiencies in embodied intelligent robots, particularly in dynamic real-time reasoning, fine physical actions, and identifying opportunities to seek human assistance. The practicality and research value of this work are commendable, as it tackles critical challenges in the field of robotics and human-robot interaction.

However, the proposal's clarity regarding its innovation over prior work is somewhat obscured. Specifically, the section on "LLM-Based Multi-Agent Systems" lacks a clear delineation of how the proposed framework differs from and improves upon existing methodologies. The paper would benefit from a more detailed exposition on the unique contributions it offers to the field.

Furthermore, the proposal is somewhat vague on the specific algorithms and methodologies that will be employed to address the identified challenges. Without a more concrete description of the approach, it is difficult to assess the feasibility of the framework. Additionally, the paper does not mention the datasets that will be used or the design philosophy behind the key modules, which are crucial for evaluating the practical implementation of the proposed system.

In conclusion, while the paper presents an intriguing concept with significant potential, it requires further elaboration on its innovative aspects and a more detailed description of its technical underpinnings. Addressing these gaps would greatly enhance the paper's contribution to the field and provide a clearer path for future research and development.

---

### Official Review · ~André_Moreira_Leal_Leonor1 · 2024-11-09
**Enhancing embodied agents' reasoning through LLM-based multi-agent collaboration for text tasks**

**Rating:** 10
**Confidence:** 4

**Review:**

This proposal reflects the aim of the research: using a multi-agent framework with large language models to enhance reasoning and task execution in embodied agents, with a special focus on text-based tasks in human-populated settings. It really represents a collaborative approach in nature and aims at solving real-world dynamic environment challenges.

---

### Official Review · ~Yuanda_Zhang1 · 2024-11-09
**Interesting question, great idea**

**Rating:** 10
**Confidence:** 4

**Review:**

The proposal presents an innovative study that aims to improve the reasoning capabilities of embodied agents for solving text-based tasks in human-populated environments. By leveraging Large Language Models (LLMs) within a multi-agent system, the authors intend to create a more effective framework for task execution and human-robot interaction, with a focus on reducing human intervention in domestic settings.

Pros:
1）The research question addressed by the proposal is both novel and intriguing, with significant practical implications for the field of robotics and embodied AI.
2）The literature review is comprehensive, covering a wide range of relevant research and demonstrating the authors' thorough understanding of the current state of the art.
3）The proposed methodology, which involves designing a multi-agent reasoning framework and conducting tests in simulated and real-world environments, is promising and has the potential to advance the capabilities of embodied agents significantly.

Cons:
1）While the methodology section outlines a clear plan for testing and evaluation, there is a lack of preliminary results or insights from any pilot experiments that would support the feasibility of the proposed approach.

---

### Official Review · ~Chenxi_Hu4 · 2024-11-11
**Promising Topic but Limited Clarity in Multi-Agent Collaboration Framework**

**Rating:** 8
**Confidence:** 4

**Review:**

The proposal offers a promising solution to enhance embodied agents’ capabilities in human-populated environments through LLM-based multi-agent collaboration. While the research objectives and methodology are well-defined, further clarification on LLM agent roles, human-robot interaction specifics, and detailed evaluation metrics is needed. But still, this project is a novel topic with the potential to have a significant impact on the real world.

---

### Official Review · ~KAI_JUN_TEH1 · 2024-11-11
**A clear problem statement and a well-defined research plan**

**Rating:** 10
**Confidence:** 4

**Review:**

This paper proposes a multi-agent reasoning framework that leverages large language models (LLMs) to enhance the capabilities of embodied agents, such as robots, in executing text-based tasks in human-populated environments like offices.
First, the article points out that in office-like environments, robots or embedded systems still face challenges such as dynamic reasoning, limited physical abilities, and human interaction. Therefore, the article proposes an LLM-based multi-agent reasoning framework to tackle the aforementioned issues. It also plans to build a simulated real-world environment to validate the effectiveness of this framework. In my view, this is a very solid piece of work.
However, my concern is that the article does not clarify the advantages of multi-agent systems, which led the authors to choose them over other systems. Thanks!

---

### Official Review · ~jin_wang30 · 2024-11-12
**A detailed and practical proposal**

**Rating:** 10
**Confidence:** 4

**Review:**

This proposal explores a multi-agent collaboration framework driven by a large language model, aiming to enhance the reasoning ability of robots to perform tasks in complex environments. The system proposed in this paper is suitable for robot assistance in text-command tasks, and improves the performance of robots in dynamic and changeable real-world scenarios through multi-agent collaboration.

Advantages:
This paper focuses on applying the reasoning ability of LLM to environments with intensive human-computer interaction, such as offices and homes. The research content is forward-looking and in line with the current development trend in the field of artificial intelligence and robotics. In addition, this paper designs a multi-agent collaboration framework that enables different agents to collaborate and divide labor in robot tasks, which has good innovation and practical value. The roles and tasks of each agent are described in detail, and the framework design is reasonable. The structure of the paper is very clear, and multi-dimensional evaluation indicators are proposed, including task completion rate, human-computer collaboration efficiency, and the degree of reduction of human intervention, which ensures the reliability of the research results.

Disadvantages:
Although the article proposes a theoretical framework, it lacks actual test or experimental data to verify its effectiveness. If it can be combined with actual experimental data and specific test results, it will be more convincing. In addition, the article lacks detailed descriptions of the specific interaction mode and data flow between LLM and multi-agents. It would be better if the details were properly enriched.
Overall, this is a very good proposal, and I look forward to the subsequent research progress.

---

### Official Review · ~Chaoqun_Yang2 · 2024-11-12
**Clear problem definition**

**Rating:** 9
**Confidence:** 4

**Review:**

**Summary:**
The proposal addresses the challenge of enhancing robots' ability to interpret and execute text-based instructions in dynamic human environments. The authors propose an LLM-based Multi-Agent Reasoning Framework to improve robots' dynamic reasoning, physical capabilities, and human interaction skills. This framework aims to enable robots to autonomously handle a variety of tasks while minimizing the need for human intervention, particularly in office settings. The proposal outlines the challenges faced by current robots, such as dynamic reasoning and limited physical abilities, and suggests a collaborative system where LLM-driven agents assist robots in processing human instructions more effectively.

**Highlights:**
1. **Comprehensive Objectives:** The research objectives are well-defined and cover the development, validation, and iterative improvement of the proposed framework, ensuring a thorough approach to solving the problem.
2. **Interdisciplinary Approach:** The proposal brings together elements of robotics, AI, and human-robot interaction, showcasing an interdisciplinary approach that is crucial for advancing embodied agents' capabilities.

**Advice:**
1. **Methodological Details:** It would be better if more details on the methodology and experimentation is provided, such as the specific baseline approaches for comparison, and the implementation details of the proposed method.
2. **The innovation of the framework:** What's the innovation of the proposed framework? How is it different from the common multi-agent cooperative framework? What idea is adopted to solve the problem to be solved in this paper?

---

### Official Review · ~Maanping_Shao1 · 2024-11-12

**Rating:** 9
**Confidence:** 3

**Review:**

The proposal, "Leveraging LLM-based Multi-Agent Collaboration to Enhance Embodied Agents' Reasoning Capabilities," presents an innovative approach to improve robotic task execution in human-populated environments. By employing a multi-agent framework with Large Language Models (LLMs), the authors aim to enhance robots’ reasoning abilities and reduce human intervention.

Strengths:

Innovative Approach: Using LLMs for multi-agent collaboration could greatly improve robots' adaptability and human interaction capabilities.
Real-World Impact: The research targets critical challenges, with potential applications in dynamic, indoor environments.
Methodology: The proposed testing in simulated and real environments, combined with iterative improvement, is well-conceived.
Weaknesses:

Technical Detail: The integration of LLMs with agents could be more clearly defined.
Evaluation Metrics: More comprehensive success metrics would benefit the evaluation.

---

### Official Review · ~Zihan_Wang7 · 2024-11-12
**Hard and Meaning Task**

**Rating:** 9
**Confidence:** 4

**Review:**

**Summary**

This research proposal aims to enhance robots' capabilities in human environments through an LLM-based multi-agent reasoning framework. Through a multi-LLM agent collaboration system, it helps robots better understand and execute text-based tasks while intelligently seeking human assistance when needed.

**Highlights**
1. Key problems to be addressed: limitations in dynamic reasoning in real-time situations, physical constraints in complex task execution, difficulties in human-robot interaction and assistance-seeking
2. Research methodology: design of multi-agent collaboration framework, testing in both simulated and real-world environments, evaluation based on task completion rates and human intervention metrics, iterative improvement based on feedback

**Advice**
1. Include more detailed metrics for measuring "human intervention minimization"
2. Define specific scenarios for testing the framework

---

### Decision · Program_Chairs · 2024-11-18

**Decision:**

Strong Accept (Long Presentation)

**Comment:**

**Enhancing Embedded Agents' Text-based Reasoning Abilities in Human Environments through LLM-based Multi-Agent Collaboration**

**2.4.1 Key Innovations**
1. Developing a collaborative framework for multi-agent systems

**2.4.2 Additional Key Information**
None

**2.4.3 Advantages**
1. Potential applications in dynamic indoor environments

**2.4.4 Areas for Improvement**
1. Provide a clearer explanation of why individual agents struggle to effectively handle real-world problems
2. Define how the collaborative framework can achieve self-improvement
3. Clearly outline the conditions and boundaries for triggering human-agent collaboration within the framework

**2.4.5 Recommendations**
1. Differentiate between "agents" and "embodied intelligence" in the descriptions
2. Review existing research in this domain
3. Investigate how traditional multi-agent research can inform and contribute to this study